# Condensing Multilingual Knowledge
# with Lightweight Language-Specific Modules

**Haoran Xu, Weiting Tan\*, Shuyue Stella Li\*, Yunmo Chen\*,**
**Benjamin Van Durme, Philipp Koehn, Kenton Murray**

Johns Hopkins University
{hxu64,wtan12,sli136,yunmo,phi,kenton}@jhu.edu

## Abstract

Incorporating language-specific (LS) modules or Mixture-of-Experts (MoE) are proven methods to boost performance in multilingual model performance, but the scalability of these approaches to hundreds of languages or experts tends to be hard to manage. We present Language-specific Matrix Synthesis (LMS), a novel method that addresses the issue. LMS utilizes parameter-efficient and lightweight modules, reducing the number of parameters while outperforming existing methods, e.g., +1.73 BLEU over Switch Transformer on OPUS-100 multilingual translation. Additionally, we introduce Fuse Distillation (FD) to condense multilingual knowledge from multiple LS modules into a single shared module, improving model inference and storage efficiency. Our approach demonstrates superior scalability and performance compared to state-of-the-art methods.[1]

## 1 Introduction

Multilingual models confer the benefit of facilitating cross-lingual learning; however, they also grapple with the issue of language interference (Conneau et al., 2020; Wang et al., 2020a; Shaham et al., 2022). Recent studies aim to alleviate negative language interference through the introduction of language-specific (LS) modules (Zhang et al., 2020; Fan et al., 2020; Zhang et al., 2021; Fan et al., 2021; Pires et al., 2023). In this setup, each language batch is processed through its designated module rather than a shared module. Although this approach is promising and barely inflates the number of FLOPs like Mixture-of-Experts (MoE) (Shazeer et al., 2017; Lepikhin et al., 2021),[2] the number of parameters becomes

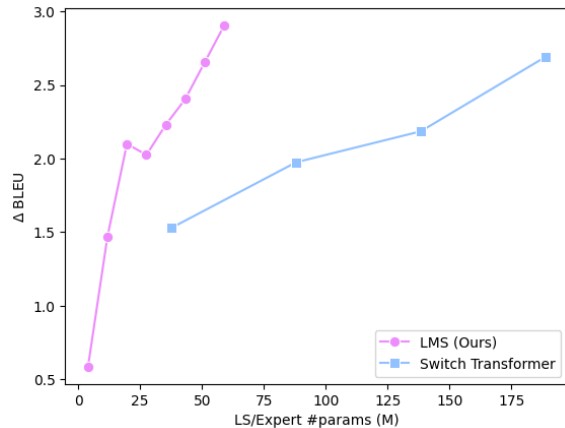

Figure 1: We show the BLEU gains between the LMS method and the Switch Transformer as the model's parameters increase in our multilingual translation ablation study. The LMS method notably outperforms the Switch Transformer with similar extra LS (expert) parameter counts, achieving comparable performance even with four to five times fewer parameters.

difficult to manage and sometimes impractical when working with a large variety of languages. This is because the fundamental element forming LS or MoE modules is typically the full-rank weight matrix derived from a densely connected layer, which causes a rapid increase in the number of parameters with a large number of languages or experts.[3]

In this paper, we first scrutinize the parameter efficiency of language-specific modules from the perspective of using fewer parameters. Consequently, a necessary question arises (*RQ1*): *can we approximate the original dense weight matrix using substantially fewer parameters?* To answer this question, we propose novel and parameter-efficient method, **Language-Specific**

---

\* Equal contribution

[1]We release our code at: https://github.com/fe1ixxu/LMS_FD.

[2]Each pass through the model utilizes only the corresponding language-specific component. The additional

computational cost may only come from communication among devices (such as ALLToALL) or gate routing.

[3]Although MoE employs a routing mechanism to keep the number of experts smaller than the number of languages, the parameter cost remains substantial.

**Matrix Synthesis** (LMS), which can achieve similar performance to switch transformer even with three to four times smaller LS parameters (as shown in Figure 1).

Then, we further investigate parameter efficiency from the perspective of knowledge density in each LS module. Given recent discoveries that the performance improvement of sparsely activated models diminishes with an increase in the number of experts (Hoffmann et al., 2022; Gao et al., 2022; Xu et al., 2023), we hypothesize that knowledge in these experts (or LS modules) is over-estimated. Hence, we propose another question (*RQ2*): *Could a single shared module encapsulate the same level of knowledge as language-specific modules?* In addressing this question, we introduce the **Fuse Distillation** (FD) method to examine the feasibility of condensing the multilingual knowledge into a single module.

Our main contributions are summarized as follows:

- We propose the parameter-efficient and lightweight LMS method, which substantially outperforms previous LS methods or MoE with fewer than or the same number of parameters, e.g., +1.73 BLEU over Switch Transformer on OPUS-100 multilingual translation.

- We introduce FD to condense multilingual knowledge from LS modules into a shared module. FD is able to use only 2M more parameters (1% increase) to achieve the 65% of performance gains from Switch Transformer which use 760M more parameters (314% increase) during inference.

- LMS and FD show strong generalization performance among multiple tasks, including multilingual machine translation (MMT) (Zhang et al., 2020), multilingual named-entity recognition (MNER) (Pan et al., 2017), and multilingual question answering (MQA) (Artetxe et al., 2020).

## 2 Lightweight LS Modules

In this section, we address *RQ1* by constructing LS modules with significantly fewer parameters.

### 2.1 Language-Specific Matrix Synthesis

Language-specific modules are typically composed of linear projections, whose weights are full-rank matrices in previous studies. We propose

the **L**anguage-specific **M**atrix **S**ynthesis (LMS) method to form low-rank matrices to approximate the full-rank ones. This is inspired by the concept of "intrinsic dimension" in pre-trained language models (Aghajanyan et al., 2021; Hu et al., 2021) and "intrinsic rank" in trainable matrices, leading to the idea that features are learned in a subspace. Specifically, as shown in Figure 2, our LS matrix is derived from the multiplication of an LS 'vertical' matrix with an LS 'flat' matrix. Formally speaking, let $W \in \mathbb{R}^{r \times c}$ be a weight matrix in the model and we want to build parallel LS matrices which have the same size. Hence, for each language $l_i$, $i \in \{1, 2, \cdots, L\}$ with $L$ being the number of languages, there exists an LS vertical matrix $W_v^{l_i} \in \mathbb{R}^{r \times d}$ and an LS flat matrix $W_f^{l_i} \in \mathbb{R}^{d \times c}$ ($d \ll \min(r, c)$) that we use to approximate the full-rank matrix. Here, we propose two synthesis methods: *language-wise* and *pair-wise synthesis*.

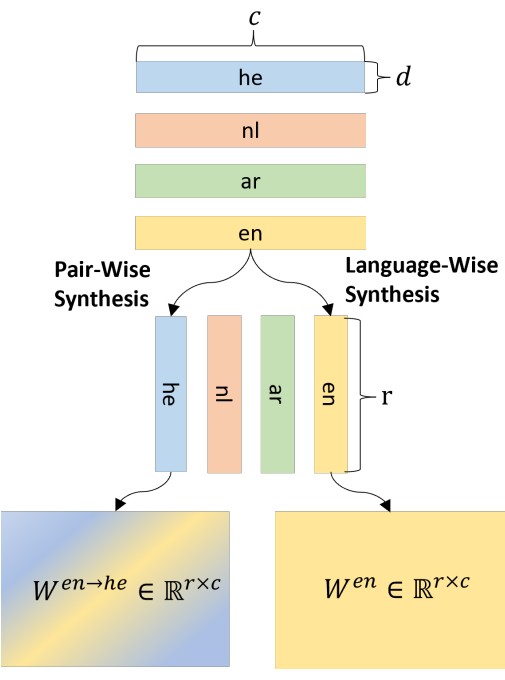

Figure 2: The difference between pair- and language-wise synthesis. Language-wise synthesis constructs a low-rank matrix using both the vertical and flat matrices derived from the same language. Conversely, pair-wise synthesis formulates the matrix by combining the vertical matrix from the source language with the flat matrix from the target language.

**Language-Wise Synthesis** Most multilingual tasks, such as conventional multilingual question-answering, are characterized by a language-monolithic nature: a single example only pertains to a single language, and examples from different

languages build the multilingual data. Under such circumstances, a naive way to assemble a language-specific matrix for a given language, $l_i$, is straightforwardly using its corresponding vertical and flat matrices, such that $W^{l_i} = W_v^{l_i} W_f^{l_i}$.

**Pair-Wise Synthesis** Cross-lingual tasks like MMT can also be accomplished using language-wise synthesis, wherein the encoder uses the source language matrix and the decoder uses the target language matrix. However, we posit that this is not the optimal strategy for MMT tasks due to the lack of learning bilingual information. Motivated by this, we introduce a pair-wise synthesis method to accommodate the bilingual context in each example in MMT. In this strategy, the language-specific matrix is a composition of the vertical matrix from the source language $l_i$ and the flat matrix from the target language $l_j$: $W^{l_i \to l_j} = W_v^{l_i} W_f^{l_j}$. The difference between the language-wise and pairwise synthesis approaches is depicted in Figure 2. In Section 5, we will demonstrate that the pair-wise synthesis approach is more effective.

After deriving a language-specific matrix, we incorporate it into the original full-rank matrix, as opposed to performing an isolated forward pass of the model like MoE and conventional LS methods. This approach stems from our hypothesis that the employment of low-rank matrices alone may not sufficiently facilitate the learning of features. Therefore, given an input $x_i$ associated with a source language $l_i$ and a target language $l_j$ ($l_i$ and $l_j$ are the same for language-monolithic tasks), our modified forward pass yields the output $x_o$:

$$x_o = (W + W^{l_i \to l_j})x_i = (W + W_v^{l_i} W_f^{l_j})x_i. \quad (1)$$

## 2.2 Where to Implement?

We primarily focus on incorporating language-specific matrices generated using the LMS method into the linear projection of each feedforward network (FFN) layer in every transformer layer. Recall from earlier that $r$ and $c$ are the number of rows and columns in the matrix, and $L$ is the number of languages. Thus, the total number of language-specific parameters added is given by $2L \cdot N \cdot d \cdot (c + r)$, where $N$ represents the number of layers. We also conduct an ablation study to examine the performance when implementing LMS in attention layers in Section 6. For initialization, we employ a random Gaussian distribution for vertical matrices and zeros for flat matrices suggested by Hu et al. (2021).

# 3 Can We Fuse Multilingual Knowledge in A Single Module?

In this section, we introduce **F**use **D**istillation (FD) and use a preliminary experiment to answer *RQ2*: whether we can condense the multilingual knowledge from language-specific modules into a single module.

## 3.1 Fuse Distillation

Let us first consider a language- (or task-) level MoE (Kudugunta et al., 2021), where we replace a single FFN layer with $L$ FFN modules. $L$ is the number of languages, as defined previously. The slight difference from the original design is we discard the routing gate and make each expert language-specific, i.e., an expert only serves batches in its corresponding language. Given recent findings that model improvements diminish with an increasing number of experts (Hoffmann et al., 2022; Gao et al., 2022; Xu et al., 2023), we hypothesize that information contained in experts is sparse and can be condensed into a shared module. To fuse knowledge from $L$ FFN layers to the shared one, we propose the following training scheme and name this method Fuse Distillation:

We first add an additional shared FFN parallel to an existing model with $L$ FFN layers as shown in Figure 3. During training, each batch undergoes two forward passes and one backward pass. In the first forward pass, the batch is processed through its language-specific FFN module; in the second pass, the batch is routed through the shared FFN. To fuse the language-specific knowledge contained within the $L$ FFN modules into the shared FFN module, a distillation loss between the outputs from the two forward passes is also incorporated:

$$\mathcal{L}_{fd} = \mathbb{KL}(g(p_l) \parallel p_s). \quad (2)$$

where $p_l$ denotes the probability output for the LS pass, and $p_s$ represents the shared pass output. The function $g(\cdot)$ signifies that gradients will not be traced back, so only the shared module learns from LS modules but LS ones do not learn from this loss. The backward pass also involves optimizing the model by minimizing the Cross-Entropy loss ($\mathbb{CE}$) between the target and predicted values (the regular training loss). Thus, the total loss is:

$$\mathcal{L} = \frac{1}{2}(\mathbb{CE}(y \parallel p_l) + \mathbb{CE}(y \parallel p_s)) + \mathcal{L}_{fd}, \quad (3)$$

where $y$ denotes gold labels.

Then, **during the inference stage, we discard the LS modules**. The model only forward passes the shared FFN for inference. To evaluate whether the shared FFN has effectively learned all LS information, we conduct a comparison between its results and those obtained via the routing through LS modules instead.

### 3.2 Preliminary Experiments

Our preliminary experiments are conducted under three settings:
**(1) Naive MMT:** A basic multilingual translation model is trained without any modifications.
**(2) FD:** This setting utilizes our proposed fuse distillation method.
**(3) FD-LS:** We train the model with the FD method, but during the inference stage, the input is processed through its language-specific FFN module instead of the shared module as the original language-level MoE did.

We carry out our experiments using the IWSLT benchmarks, focusing on the many-to-many translation model paradigm. Following Lin et al. (2021); Xu et al. (2022), we collect 8 English-centric language pairs from the IWSLT'14 dataset, with sizes ranging from 89K to 169K sentences. We train all methods with the same number of steps and leave detailed training settings in Appendix A. We report sacreBLEU scores (Papineni et al., 2002; Post, 2018) with the FLORES-200 tokenizer (NLLB Team et al., 2022).

### 3.3 Results and Analysis

Overview results of these 4 settings are shown in Table 1. The reported scores are the average of both xx→en and en→xx directions. As anticipated, after applying language-specific modules for each FFN layer, FD-LS has considerable enhancements over the naive MMT (+1.50 BLEU gains). Importantly, after discarding LS modules, FD only performs slightly worse than FD-LS (+1.17 vs. +1.50) with much fewer parameters for inference (48M vs. 149M). This observation underscores the feasibility of condensing multilingual knowledge into a single FFN module, thereby reducing the need of a large number of LS parameters for inference.

## 4 Combining LMS and FD

We have shown the success of multilingual information condensation by fuse distillation. We are interested in further reducing the parameters

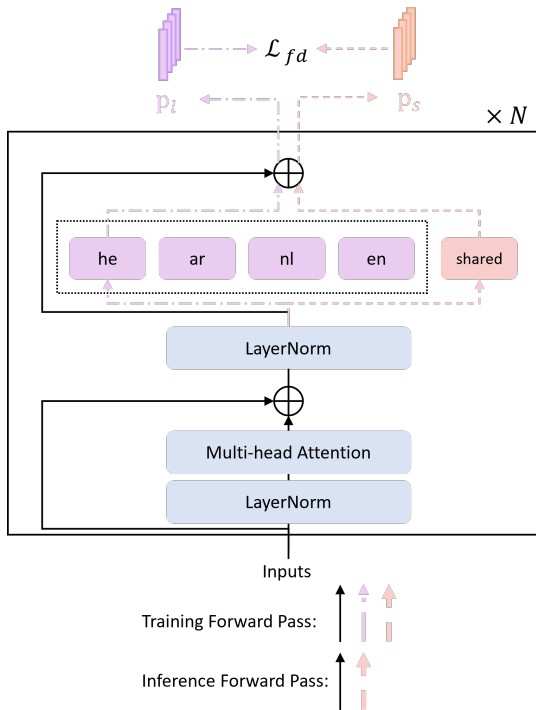

Figure 3: We utilize a language-level MoE architecture to verify the feasibility of fusing multilingual knowledge from all language-specific modules into a single shared module. During training, each batch goes through the LS module in the first forward pass and goes through the shared module in the second pass. Then, we conduct distillation between two outputs to condense the knowledge into the shared module. For inference, we discard the LS module and only use the shared module.

needed by utilizing the language-specific matrix synthesis method during inference, so we then attempt to incorporate the FD method within LMS. Similar to Section 3.1, apart from the LS vertical and flat matrices, we introduce shared vertical and flat matrices, denoted as $W_v^{\text{shared}}$ and $W_f^{\text{shared}}$, respectively. To employ the fuse distillation method, each batch is required to undergo two forward passes. The initial pass navigates through the LS matrix $W + W_v^{l_i} W_f^{l_j}$, while the subsequent pass traverses the shared matrix $W + W_v^{\text{shared}} W_f^{\text{shared}}$. These two passes generate two respective outputs, $p_l$ and $p_s$. Given the common parameter $W$ shared across both paths, we utilize symmetric KL divergence (Jiang et al., 2020) for distillation, as opposed to the traditional KL divergence:

$$\mathcal{L}'_{fd} = \frac{1}{2}(\mathbb{KL}(p_l \parallel p_s) + \mathbb{KL}(p_s \parallel p_l)). \quad (4)$$

Thus, the backward pass optimizes both the standard prediction loss and the fuse distillation

| Methods | ar | de | es | fa | he | it | nl | pl | avg. | #params | |
|---|---|---|---|---|---|---|---|---|---|---|---|
| | | | | | | | | | | Training | Inference |
| Naive MMT | 25.03 | 32.59 | 39.98 | 18.76 | 33.39 | 34.00 | 36.71 | 22.37 | 30.35 | 48M | 48M |
| FD | +1.01 | +1.15 | +1.43 | +0.64 | +1.44 | +1.19 | +1.22 | +1.22 | +1.17 | 161M | 48M |
| FD-LS | **+1.30** | **+1.45** | **+1.72** | **+0.77** | **+2.08** | **+1.48** | **+1.41** | **+1.73** | **+1.50** | 161M | 149M |

Table 1: Average BLEU on IWSLT'14 many-to-many translation. Our proposed FD is able to fuse the majority of knowledge into a single module (+1.17 vs. +1.50) with the same parameters as the naive model during inference.

Figure 4: Suppose we incorporate additional language-specific (LS) linear projections into a layer. We compare the space complexity of the extra LS parameters (or experts) needed across all methods for both training and inference phases. Let's denote $L = 15$ as the number of languages, $r = 4096$ as the output dimension, $c = 1024$ as the input dimension, $E = 8$ represents the number of experts for Mixture-of-Experts (MoE), and $d = 32$ signifies the rank for low-rank matrices. The number adjacent to the dashed line is the number of parameters calculated based on the given sample numbers. In this case, one can observe that the Language-Specific Matrix Synthesis (LMS) requires a significantly lower quantity of LS parameters compared to other methods during training, and fuse distillation (FD) demands a substantially reduced number of additional parameters during the inference stage.

loss.

In Figure 4, we provide a comprehensive comparison of space complexity for generating extra LS (or expert) modules, among conventional LS modules, Mixture-of-Experts, and our proposed methods. Notably, our methods demonstrate substantial reductions in parameter usage during both training and inference.

## 5 Experiments

We evaluate our LMS and LMS+FD methods using three tasks: MMT, MNER, and MQA. Similar to Section 3.2, we have two routing options for the LMS+FD method during inference time: 1) evaluating the model by passing the shared route (denoted as LMS+FD-Share, the default setting), or 2) passing the language-specific module (denoted as LMS+FD-LS). We present results for both routes to show the performance difference between using the condensed module and the original LS modules. Considering the computational cost for MMT, we run all methods once with the same random seed. For the other two tasks, we run experiments with 3 different random seeds and report the average scores. For ease of implementation, we build homogeneous

batches (i.e., a batch only containing sentences in one language or one language direction) and only activate the corresponding LS module.[4]

### 5.1 Baselines

We compare our approaches against two strong baselines that incorporate additional parameters to mitigate language interference.

**CLSR:** The first baseline is Conditional Language-Specific Routing (CLSR) (Zhang et al., 2021), which employs LS linear projections following FFN or attention layer. Following their best settings, we set the budget $p = 0.3$ for LS routing. The original setting used shared LS projections across all encoder or decoder sublayers. We also consider a non-shared version, where each sublayer has its own LS projection, and denote it as CLSR*.

**Switch Transformer:** We also consider Switch Transformer (Fedus et al., 2021) as the second strong baseline, which uses similar FLOPs as our methods.[5] We use 16 experts for every two layers

---

[4]This does not apply to Switch Transformer.

[5]The design of the Switch Transformer, which employs top-1 routing, bears similarity to our model in that it processes

| Methods | ar | de | es | fa | he | it | nl | pl | avg. | #params | |
|---|---|---|---|---|---|---|---|---|---|---|---|
| | | | | | | | | | | Training | Inference |
| Naive MMT | 25.03 | 32.59 | 39.98 | 18.76 | 33.39 | 34.00 | 36.71 | 22.37 | 30.35 | 48M | 48M |
| Switch Transformer | +0.28 | +0.40 | +0.45 | +0.04 | +0.60 | +0.59 | +0.34 | +0.67 | +0.42 | 149M | 149M |
| CLSR | +0.00 | +0.48 | +0.51 | -0.23 | +0.31 | +0.50 | +0.42 | +0.30 | +0.28 | 53M | 53M |
| CLSR* | +0.66 | +0.87 | +1.16 | +0.53 | +0.99 | +1.00 | +0.87 | +0.94 | +0.88 | 105M | 105M |
| LMS, lang-wise | +0.48 | +0.53 | +0.88 | +0.83 | +0.86 | +0.91 | +0.81 | +0.91 | +0.78 | 58M | 58M |
| LMS | +0.87 | +1.08 | +1.04 | +0.62 | +1.37 | +1.20 | +1.04 | +1.16 | +1.05 | 58M | 58M |
| LMS+FD-Share | +0.82 | +0.93 | +1.06 | +0.34 | +1.23 | +0.92 | +0.87 | +0.83 | +0.88 | 60M | 49M |
| LMS+FD-LS | **+1.23** | **+1.34** | **+1.44** | **+0.77** | **+1.51** | **+1.36** | **+1.24** | **+1.15** | **+1.26** | 60M | 58M |

Table 2: Overall BLEU results of on IWSLT'14 many-to-many translation. LMS outperforms all baselines. At inference, LMS+FD-Share utilizes extra 1M parameters to exceed baselines that enlarge the model size 2 or 3 times.

| Methods | en→xx | | | | | xx→en | | | | | #params | |
|---|---|---|---|---|---|---|---|---|---|---|---|---|
| | high | med | low | all | WR (%) | high | med | low | all | WR (%) | Training | Inference |
| Naive MMT | 23.89 | 31.17 | 29.76 | 27.37 | - | 29.40 | 31.85 | 31.49 | 30.60 | - | 242M | 242M |
| Switch Transformer | +1.87 | +3.29 | **+3.51** | +2.66 | **100** | +1.18 | +1.15 | -0.31 | +0.84 | 83 | 1002M | 1002M |
| CLSR | +0.02 | +0.00 | +0.01 | +0.02 | 52 | +1.33 | +2.00 | +2.71 | +1.83 | 91 | 443M | 443M |
| LMS, lang-wise, $d=64$ | +2.12 | +2.28 | +1.77 | +2.09 | 95 | +1.85 | +2.34 | +2.30 | +2.09 | 94 | 989M | 989M |
| LMS, $d=64$ | **+3.60** | **+3.82** | +3.32 | **+3.60** | 99 | **+2.75** | **+3.74** | **+4.16** | **+3.35** | 95 | 989M | 989M |
| LMS+FD-Share, $d=64$ | +0.49 | +0.75 | +1.29 | +0.74 | 88 | +0.64 | +1.52 | +2.08 | +1.22 | 98 | 996M | 250M |
| LMS+FD-LS, $d=64$ | +1.72 | +2.03 | +2.60 | +2.01 | **100** | +1.64 | +2.82 | +4.03 | +2.52 | 99 | 996M | 996M |
| LMS, $d=16$ | +2.45 | +2.62 | +2.56 | +2.53 | 99 | +1.75 | +2.68 | +3.40 | +2.39 | 96 | 429M | 429M |
| LMS+FD-Share, $d=16$ | +0.54 | +1.13 | +2.20 | +1.09 | 94 | +0.81 | +1.26 | +1.85 | +1.17 | 94 | 431M | 244M |
| LMS+FD-LS, $d=16$ | +1.28 | +1.84 | +2.74 | +1.77 | **100** | +1.35 | +2.25 | +3.53 | +2.10 | **100** | 431M | 431M |
| LMS, $d=4$ | +1.72 | +2.05 | +2.31 | +1.95 | 99 | +1.33 | +1.80 | +1.71 | +1.55 | 93 | 289M | 289M |

Table 3: BLEU scores on OPUS-100 many-to-many translation. LMS with $d=64$ outperforms all baselines on average. LMS+FD-Share with $d=16$ uses 1% more parameters, and achieves 65% BLEU gains averaged by all directions, compared to the Switch Transformer which uses 314% more parameters.

with a gate balance loss with a weight of 0.01.

## 5.2 Multilingual Machine Translation

**Data and Training settings** We concentrate on the many-to-many translation setting, with results reported from two benchmarks. The first is the English-centric IWSLT'14 dataset, as aforementioned in Section 3.2. Additionally, we examine the OPUS-100 dataset (Zhang et al., 2020), which encompasses 100 languages in total, including 94 development/test language pairs. We preprocess the data by sentencepiece (Kudo and Richardson, 2018), establishing a vocabulary size of 32K for the IWSLT'14 dataset and 64K for the OPUS-100 dataset. We utilize transformer$_{small}$ and transformer$_{big}$ for IWSLT'14 and OPUS-100, respectively. We fix the training steps for all methods for a fair comparison. For IWSLT'14, we use $d=32$ as the rank for low-rank matrices. For OPUS-100, we consider three settings: (i) $d=64$ to match the parameter size of the Switch Transformer, (ii) $d=16$ to match the parameter size of CLSR, and (iii) $d=4$ for very

through a single module in each expert layer.

lightweight LS model construction. The default LMS setting for MMT tasks is pair-wise unless otherwise specified. We discuss more training details in Appendix A.

**Evaluation** We report results in terms of sacreBLEU (Post, 2018), tokenized by FLORES-200 tokenizer (NLLB Team et al., 2022), and win ratio (WR) (Zhang et al., 2020) which is the proportion of language pairs on which our method beats the baseline. For IWSLT'14, we report the scores averaged by xx→en and en→xx directions. For OPUS-100, we split the 94 test language pairs into three groups based on their training data size suggested by Zhang et al. (2020): high-resource (> 0.9M, 45 languages), low-resource (< 0.1M, 21 languages) and medium-resource (others, 28 languages), and report the averaged scores in each category. We use beam search with a width of 5 and use a length penalty of 1.

**LMS performance: Light and Effective LS Module** The primary results for IWSLT'14 and OPUS-100 are presented in Table 2 and Table 3, respectively. In the IWSLT'14 dataset, LMS

significantly surpasses both the Switch Transformer and CLSR, despite having considerably fewer parameters. For OPUS-100, our methods and the baselines are evaluated with approximately equal extra parameters (e.g., 1002M in the Switch Transformer and 989M in LMS with $d = 64$). Compared with the gains from Switch transformer (+2.66 for en→xx and +0.84 for xx→en), our pair-wise LMS method achieves substantially higher gains (+3.60 and +3.35). Similarly, our LMS method also outperforms CLSR (+0.02 and +1.83) with a comparable number of extra parameters. These results show the strong parameter efficiency of LMS for the MMT tasks. With merely 47M parameters ($d = 4$), our LMS method matches the Switch Transformer's performance for en→xx and the CLSR's performance for xx→en.

**Language-Wise or Pair-Wise?** We compare language- and pair-wise synthesis in both IWSLT'14 and OPUS-100 ($d = 64$) datasets. On average, pair-wise synthesis outperforms language-wise synthesis by 0.27 BLEU points on IWSLT'14 (+1.05 vs. +0.78). Moreover, the pair-wise method (+3.60 and +3.35) also shows superior performance on the OPUS-100 dataset compared with the language-wise one (+2.09 and + 2.09). Notably, pair-wise synthesis with $d = 16$ surpassed the performance of language-wise synthesis with $d = 64$, even though the latter has 4 times more extra parameters. Hence, this discovery strongly advocates for the use of pair-wise synthesis over the language-wise approach.

**FD performance: Can FD Fuse 95 Languages?** On the IWSLT'14 8-language MMT dataset, we observe negligible differences between LMS and LMS+FD (+1.05 vs. +0.88), suggesting successful condensation of information from various language-specific modules into the shared module. In the 95-language (94 languages plus English) scenario of OPUS-100, FD with a dimensionality of 16 utilizes only an additional 2M parameters (less than 1% increase compared to the 242M naive model) to attain 65% of the performance improvements from Switch Transformer (+1.13 vs. +1.75 on average), which requires 760M additional parameters (a 314% increase). While FD may not condense all multilingual information due to restricted parameter capacity, its parameter efficiency is commendable.

| Methods | Sampled Language | | avg. | WR (%) | #params | |
| | qu | vi | | | Tra. | Inf. |
| --- | --- | --- | --- | --- | --- | --- |
| Naive MNER | 76.79 | 92.60 | 89.20 | - | 270M | 270M |
| LMS | +3.61 | +0.28 | +0.55 | 96 | 340M | 340M |
| LMS+FD-Share | +3.22 | +0.45 | +0.33 | 88 | 343M | 273M |
| LMS+FD-LS | **+3.96** | **+0.57** | **+0.67** | **100** | 343M | 340M |

Table 4: The overall MNER results (F1 score) between baseline and our three proposed methods.

## 5.3 Multilingual Named-Entity Recognition

**Data and Settings** We evaluate our methods on Wikiann Named-Entity Recognition (Pan et al., 2017) dataset. We randomly select 24 languages to conduct experiments. The model architecture is based on pre-trained XLM-R$_{base}$, attached with a feed-forward token-level classifier. We set the dropout rate as 0.1 and run 20 epochs for all methods. We set $d = 32$ for low-rank matrices and report F1 scores.

**Results** The overall results are shown in Table 4. When applying LMS to each FFN layer for 24 languages, the model size increases by only 70M, while yielding a 0.55 F1 improvement. After implementing LMS+FD, the performance improves by 0.67 with the LS route and achieves a 0.33 gain with the shared route, which requires only an additional 3M parameters. Full results are shown in Appendix B.

## 5.4 Multilingual Question Answering

**Data and Settings** We pick 6 languages from TyDiQA (Typologically Diverse Question Answering)-Gold Passage to conduct the MQA experiments (Artetxe et al., 2020). Following Xu and Murray (2022), the representations of subwords in XLM-R$_{base}$ are input to a span classification head; a linear layer computing the answer's start and end. We set $d = 32$ for low-rank matrices, dropout rate = 0.1, and run 20 epochs.

**Results** The overall results are shown in Table 5. Upon the application of LMS and LMS+FD, all methods exhibit improved performance with a slight increase in parameters. Notably, LMS+FD-Share outperforms LMS+FD-LS. This suggests that FD may be more effective in fusing knowledge when the number of languages is relatively small. Full results are shown in Appendix C.

| Methods | Sampled Language | | avg. | WR (%) | #params | |
|---|---|---|---|---|---|---|
| | bn | sw | | | Tra. | Inf. |
| Naive MQA | 77.69 | 80.97 | 75.31 | - | 270M | 270M |
| LMS | -0.59 | **+0.93** | +0.58 | 50 | 287M | 287M |
| LMS+FD-Share | **+1.39** | +0.32 | **+1.22** | **100** | 290M | 273M |
| LMS+FD-LS | +1.26 | +0.38 | +1.15 | **100** | 290M | 287M |

Table 5: The overall MQA results (F1 score) between baseline and our three proposed methods.

| Methods | avg. BLEU | WR (%) | #params |
|---|---|---|---|
| Naive MMT | 28.05 | - | 61M |
| LMS, ffn only (default) | +2.10 | **100** | 80M |
| LMS, att only | +1.32 | **100** | 77M |
| LMS, att+ffn | **+2.14** | **100** | 96M |

Table 6: The average BLEU gains with three different LMS designs with a fixed rank $d = 20$.

# 6 Ablation Study

## 6.1 Is LMS Parameter-Efficient?

Here, we examine the parameter efficiency of the LMS method, i.e., whether an increase in extra parameters yields a proportional enhancement in model performance. We conduct experiments with $d$ ranging from 4 to 60 in increments of 8 to observe the resulting performance variations. For comparison, we examine the Switch Transformer with 4, 8, 12, 16 experts to assess its parameter efficiency. We focus on the MMT task using the OPUS-100 dataset. Due to computational demands, we limit experiments to randomly selected 15 languages from OPUS-100, designated as OPUS-15. We leave training details in Appendix D.

We report the average BLEU gains over all translation directions in Figure 1. The plot reveals that the LMS curve is steeper compared to that of the Switch Transformer, indicating a higher parameter efficiency for our method, i.e., it achieves greater model performance with fewer additional parameters. Compared with a 16-expert Switch Transformer, LMS with $d = 52$ yields similar performance by using 3.7 times smaller parameters (51M vs. 189M). Numeric results are in Appendix E.

## 6.2 Applying LMS to The Attention Layer

In our default design, the LMS is solely applied to FFN layers. We are interested in assessing the potential benefits of extending LMS to the attention layer (in each K, Q, V, output projection). We consider three model variants: (1) LMS applied only to FFN layers (default design), (2) LMS applied only to the attention layers, and (3) LMS applied to both FFN and attention layers. We conduct experiments on OPUS-15, with a fixed rank value of $d = 20$.

We show the averaged BLEU of all translation directions of the three designs in Table 6. LMS applied only to attention layers yields inferior performance compared to LMS applied only to FFN layers with a similar number of extra

parameters. Moreover, applying LMS to both FFN and attention layers results in a marginal improvement over its application solely to FFN layers. This outcome suggests that LS information is primarily situated in FFN layers, aligning with the previous findings of Wang et al. (2020b).

# 7 Related Work

**Language-Specific Modules** To mitigate language interference, previous studies incorporate language-specific modules into models, such as additional language-aware linear projections (Zhang et al., 2020; Fan et al., 2020; Zhang et al., 2021; Fan et al., 2021), LS layer normalization (Zhang et al., 2020). Feed-Forward Networks (Kwon and Chung, 2023), or even entire language-dependent transformer layers (Escolano et al., 2021; Wang and Zhang, 2022; Pires et al., 2023). Similar to LS modules, Mixture-of-Experts (MoE) are also able to reduce language interference (Shazeer et al., 2017; Lepikhin et al., 2021; Fedus et al., 2021; Xu et al., 2023). However, the parameter count of LS (or expert) drastically increases when scaling to numerous languages. Zhang et al. (2021) address this issue by sharing all LS modules across all encoder or decoder layers. However, this does not fundamentally resolve the problem, given that the complexity of constructing LS modules remains unaltered and that different layers may need to learn varying types of LS information.

**Lightweight Modules** Our proposed techniques draw inspiration from another research line, lightweight fine-tuning, wherein the model undergoes fine-tuning on a parameter subset significantly smaller than that of the original model, such as prefix tuning (Li and Liang, 2021), prompt tuning (Lester et al., 2021), multitask prompt tuning (Wang et al., 2023), LoRA (Hu et al., 2021). In the multilingual machine translation setting, previous studies use language-pair adapters (Bapna and Firat, 2019) to fine-tune a specific

direction. This approach also extends to language-wise adapters (Philip et al., 2020), language-family adapters (Chronopoulou et al., 2023), hyper-adapters (Baziotis et al., 2022) to facilitate the cross-lingual learning. In light of the efficient lightweight modules, we propose LMS to help LS modules scale to hundreds of languages.

# 8   Conclusion

The construction of language-specific modules (or experts) using full-rank matrices tends to be parameter-intensive and inefficient, especially as the number of languages (or experts) increases. To address this, we have introduced the Language-Specific Matrix Synthesis (LMS) method that approximates the original full-rank matrix. Notably, pair-wise synthesis, a variant of the LMS methods, exhibits commendable performance in MMT tasks. Further, we have proposed the Fuse Distillation (FD) approach to condense multilingual information into a shared module, thereby further diminishing parameter requirements during inference. Our methods outperform CLSR and Switch Transformer in MMT tasks and also demonstrate their effectiveness in MNER and MQA tasks.

# Limitations

One limitation of our LMS method is that it necessitates the construction of homogeneous batches, i.e., batches containing sentences exclusively in one language or language direction. However, this limitation could potentially be addressed by implementing ALLToALL communications amongst devices, a strategy that is already widely employed in Mixture of Experts (MoE) models (Lepikhin et al., 2021), which is a topic we intend to explore in future research. In each forward pass of an FFN layer, we need an additional step to multiply two small matrices, creating the low-rank large matrix. The additional cost of this operation is negligible, as the computational complexity of the FLOPs/tok for a Feedforward linear projection, given an input dimension $c$ and output dimension $r$, is $\mathcal{O}(r \cdot c)$, while the complexity for constructing the low-rank matrix with rank $d$ is $\mathcal{O}(d \cdot (r + c))$. For example, in our ablation study, when $r = 2048$, $c = 512$, and $d = 20$, the difference in computational load can be $\frac{2048 \times 512}{20 \times (512 + 2048)} \approx 20$ times less. In terms of actual training time, no significant

differences were observed; the discrepancy was less than 1 second per 100 updates. Additionally, a potentially effective strategy to enhance multilingual information encapsulation in FD could involve using a larger shared module relative to other lightweight LS modules. This could be an intriguing avenue for future research.

# Acknowledgements

We thank anonymous reviewers for their insightful feedback. We also extend our gratitude to Lingfeng Shen, Hieu Hoang, Young Jin Kim, Hany Hassan Awadalla, Stephen Rawls, and Amr Sharaf for their valuable suggestions. This work was supported in part by IARPA BETTER (#2019-19051600005). The views and conclusions contained in this work are those of the authors and should not be interpreted as necessarily representing the official policies, either expressed or implied, or endorsements of ODNI, IARPA, or the U.S. Government. The U.S. Government is authorized to reproduce and distribute reprints for governmental purposes notwithstanding any copyright annotation therein. This work is also supported in part by an Amazon Initiative for Artificial Intelligence (AI2AI) Faculty Research Award.

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

## A  Training Details for IWSLT'14 and OPUS-100

To balance the training data, we also over-sample low-resource languages with a temperature of $T = 5$ (Aharoni et al., 2019) for the OPUS-100 data and $T = 2$ for the IWSLT'14 data. We preprocess the data by sentencepiece (Kudo and Richardson, 2018), establishing a vocabulary size of 32K for the IWSLT'14 dataset and 64K for the OPUS-100 dataset. We pre-pend a special language id symbol at the beginning of the source sentence to indicate the target language. We build homogeneous batches (i.e., a batch only containing sentences in one language direction) and only activate the corresponding language-specific matrix. We set the dropout rate as 0.1 for both datasets. For the IWSLT'14 dataset, we fix the training steps at 150K with 8K warm-up steps for all methods, with a batch size of 4096 tokens. For OPUS, we fix the training steps at 100K with 8K warm-up steps for all methods, with a batch size of 4096 tokens but accumulating gradients 4 times. We train all models on 4 RTX 6000 GPUs. For the IWSLT'14 dataset, we employ the transformer$_{small}$ model (with an FFN dimension of 1024 and an embedding dimension of 512), while the transformer$_{big}$ model (with an FFN dimension of 4096 and an embedding dimension of 1024) is utilized for training the OPUS-100 dataset. The maximum learning rate is 0.0005. The optimizer is Adam (Kingma and Ba, 2014) with inverse_sqrt learning rate scheduler and weight decay of 0. We use beam search with a width of 5 and use a length penalty of 1.

## B  Full Results for MNER

We show the full results of MNER in Table 7.

## C  Full Results for MQA

We show the full results of MQA in Table 8.

## D  Training Details for The Ablation Study

We randomly pick 15 languages from the OPUS-100 data to build a smaller 15-language data (OPUS-15) for the ablation study: eu, pt, bg, sk, zh, sl, de, hr, nb, ga, rw, as, fy, mr, se. We conduct the ablation study under the many-to-many translation settings. To balance the training data, we sample the data with a temperature of $T = 5$.

We preprocess the data by sentencepiece (Kudo and Richardson, 2018), establishing a vocabulary size of 32K vocabulary. we fix the training steps at 50K with 8K warm-up steps for all methods, with a batch size of 4096 tokens. We employ the transformer$_{base}$ model (with an FFN dimension of 2048 and an embedding dimension of 512) for training the OPUS-15 dataset. The other settings are the same as Appendix A.

## E  Numeric Results for The Ablation Study

Figure 1 shows the averaged BLEU over all directions. Here, We show the detailed numeric results in Figure 9.

| Methods | az | pt | ms | af | kk | ar | qu | te | vi | my | tl | fr | hi |
|---|---|---|---|---|---|---|---|---|---|---|---|---|---|
| Naive NER | 90.12 | 92.56 | 94.7 | 91.59 | 88.25 | 89.64 | 76.79 | 82.42 | 92.60 | 73.22 | 96.65 | 90.47 | 90.63 |
| LMS | 90.47 | 92.76 | 94.87 | 92.95 | **88.45** | 89.62 | 80.4 | 83.15 | 92.88 | **75.92** | **97.00** | 90.69 | 90.87 |
| LMS-FD-Share | 90.67 | 92.79 | 94.91 | 92.29 | 87.98 | 89.74 | 80.01 | 82.61 | 93.05 | 73.18 | 96.84 | 90.61 | 91.24 |
| LMS-FD-LS | **90.90** | **93.15** | **95.13** | **93.05** | 88.25 | **89.87** | 80.75 | **83.33** | **93.17** | 74.04 | 96.94 | **90.78** | **91.54** |

| | ro | eu | tr | zh | et | hu | nl | id | el | he | en | avg. | WR (%) |
|---|---|---|---|---|---|---|---|---|---|---|---|---|---|
| Naive NER | 94.90 | 92.17 | 93.49 | 77.26 | 92.06 | 93.24 | 92.18 | 93.64 | 92.01 | 86.23 | 83.97 | 89.20 | - |
| LMS | 95.01 | 92.42 | 93.75 | 77.32 | **92.71** | 93.56 | 92.46 | 93.84 | 92.07 | 86.59 | 84.20 | 89.75 | 96% |
| LMS-FD-Share | 94.88 | 92.31 | 93.65 | 77.78 | 92.39 | 93.40 | 92.41 | 93.79 | 92.07 | 85.67 | 84.33 | 89.53 | 88 |
| LMS-FD-LS | **95.03** | **92.63** | **93.83** | **77.99** | 92.67 | **93.75** | **92.67** | **94.02** | **92.22** | **86.88** | **84.35** | **89.87** | **100%** |

Table 7: Full results for the NMER task. We report F1 scores.

| Methods | bn | en | fi | id | ko | sw | avg. |
|---|---|---|---|---|---|---|---|
| Naive MQA | 77.69 | 70.36 | 78.26 | 83.00 | 61.60 | 80.97 | 75.31 |
| LMS | 77.1 | 71.7 | 78.18 | 82.76 | **63.70** | **81.90** | 75.89 |
| LMS+FD-LS | 78.95 | **73.47** | 78.80 | 84.27 | 61.90 | 81.35 | 76.46 |
| LMS+FD-Share | **79.08** | 73.44 | **78.86** | **84.34** | 62.15 | 81.29 | **76.53** |

Table 8: Full results for the MQA task. We report F1 scores.

| Methods | en→xx | | | | | xx→en | | | | | extra #params |
|---|---|---|---|---|---|---|---|---|---|---|---|
| | high | med | low | all | WR (%) | high | med | low | all | WR (%) | Training |
| Naive MMT | 20.94 | 42.3 | 22.72 | 26.99 | - | 25.45 | 37.25 | 27.95 | 29.1 | - | - |
| Switch Transformer, $E = 4$ | 21.94 | 45.00 | 25.76 | 28.85 | **100** | 26.21 | 39.35 | 29.12 | 30.30 | **100** | 38M |
| Switch Transformer, $E = 8$ | 22.36 | 45.11 | 27.47 | 29.45 | **100** | 26.37 | 40.02 | 29.26 | 30.59 | 93 | 88M |
| Switch Transformer, $E = 12$ | 22.66 | 45.50 | 27.19 | 29.65 | **100** | 26.52 | 40.32 | 29.55 | 30.81 | **100** | 138M |
| Switch Transformer, $E = 16$ | 23.05 | 46.25 | 28.61 | 30.35 | **100** | 26.82 | 40.33 | 30.31 | 31.12 | **100** | 189M |
| LMS, $d = 4$ | 21.61 | 40.55 | 24.24 | 27.19 | 87 | 26.16 | 38.52 | 29.21 | 30.07 | **100** | 4M |
| LMS, $d = 12$ | 22.20 | 44.10 | 25.12 | 28.63 | **100** | 26.56 | 39.40 | 28.65 | 30.40 | **100** | 12M |
| LMS, $d = 20$ | 22.57 | 45.19 | 25.85 | 29.26 | **100** | 26.86 | 39.89 | 30.34 | 31.03 | **100** | 20M |
| LMS, $d = 28$ | 22.82 | 43.56 | 26.13 | 29.01 | 93 | 27.07 | 39.88 | 30.27 | 31.13 | **100** | 28M |
| LMS, $d = 36$ | 23.10 | 43.89 | 26.3 | 29.28 | 93 | 27.24 | 40.07 | 30.31 | 31.27 | **100** | 36M |
| LMS, $d = 44$ | 23.32 | 43.61 | 26.52 | 29.37 | 93 | 27.30 | 40.53 | 30.81 | 31.53 | **100** | 43M |
| LMS, $d = 52$ | 23.36 | 45.05 | 26.64 | 29.80 | 93 | 27.36 | 40.75 | 30.72 | 31.60 | **100** | 51M |
| LMS, $d = 60$ | **23.50** | **45.63** | **26.94** | **30.09** | **100** | **27.51** | **40.88** | **31.20** | **31.81** | **100** | 59M |

Table 9: The numeric results for the Figure 1.