# OpenReview forum: "Condensing Multilingual Knowledge with Lightweight Language-Specific Modules"
_EMNLP/2023/Conference — EMNLP 2023 Main_

### Official Review · Reviewer_LVVn · 2023-07-26

**Typos Grammar Style And Presentation Improvements:** line 40
**Soundness:** 4

**Excitement:**

4: Strong: This paper deepens the understanding of some phenomenon or lowers the barriers to an existing research direction.

**Paper Topic And Main Contributions:**

This paper addresses the problem of "efficient multilingual models". First, it proposes a parameter efficient alternative to language adapters, called Language Specific Matrxi Synthesis (LMS). The idea of LMS is similar to LoRA adapters: the weight matrix is decomposed into a product of 2 low-rank (language-specific) matrices.  LMS can be either language-specific (eg. for English both matrices would be English-specific), or language-pair specific (eg. en-fr would be a composition of English and French matrices). Another technique proposed in this work is Fuse distillation: it aims at distilling all the knowledge from language-specific modules into a single shared module. This is done by bringing representations of shared module close to the repreresentations done by language-specific modules, everythin is trained jointly. At inference, one can either discard language-specific module and only use shared module (LMS-FD-shared) or discard shared module and use only language-specific modules at inference (LMS-FD-LS).
Experiments are performed with multlingual NMT (IWSLT or OPUS-100 datasets), and multilingual NER and multilingual Question Answering. For translation task language-pair specific LMS outperforms Language-specific LMS. Interestingly LMS-FD-Shared also outperforms Naive NMT (but performs worse than LMS or LMS-FD-LS). This is an interesting result to me, given that in the end, once language specific layers are discarded, the model is not very different from the original Naive NMT baseline in terms of parameter count/architecture. But it looks like that such a fused distillation does help , probably acting as a regularizer during multilingual training.

**Questions For The Authors:**

- I am not sure I understand the difference between the LMS and LMS-FD-LS results. If I understand correctly LMS-FD-LS settings, you add KL-divergence loss that tries to bring close the output of LMS layers and new shared layer. However gradient updates from such loss are only applied to shared layer and not LMS layers. Therefore, according to my understanding, this should start from existing LMS model where you add shared layer (randomly initialized), and only shared layer is updated further with distillation procedure defined in the paper. However, according to the results this is not the case, since the results reported for LMS are different from the results LMS-FD-LS. Does this mean that you do not initatialize model LMS-FD-LS from LMS, but train it from scratch? Could you please bring more training details for these different settings?

- I do not understand what does WR(%) refers to in tables 3 and 7

- I am not sure I understand how do you compute training/inference parameters: is it the amount of total parameters you load in memory? In my understanding, when you know which language/laguage pair you are dealing with, one would only need to load language-specific modules for those languages, and therefore inference  (for specified language) should require less parameters compared to total amount of training parameters?

- I think it would be useful to report total amount of flops in training and inference. That would bring additinal axis of "efficiency" comparison between different models

- it would be interesting to report full results decomposed per language pair somewhere (in the appendix, or on separate document) for better

**Reasons To Accept:**

-  Both techniques proposed in the paper seem interesting. I like the idea of LMS, and language-pair LMS looks an interesting.
- Idea of the fused distillation is very interesting as well. Even though I am not fully convinced by the LMS-FD-Shared results, the fact that it outperforms NMT-Naive model is very interesting, and I believe different directions could be explored to further improve the results

**Reasons To Reject:**

- The training details of different variants are not fully documented and would harm reproducibility , eg.
1) do you strat from pretrained model and add LMS layers on top of it or do you train everythin from scratch?
2) Do LMS-FD-*  models start from LMS model or do you train LS and shared modules simultaneously?

- I do not have full understanding about how do authors compute inference parameters. Eg. we could distinguish the case of language-specifc from multilingual inference. In the case when we know the language of interest (mandatory when we rely on LMS) we could load only the parameters specific to this language(s) and it would mean that we would have much less parameters for inference than for training: thus in my understanding LMS-FD-LS should have the same amount of inference parameters as LMS-FD-Share . Similarly, CLSR should only activate language-specific modules, and have less parameters at inference that at training. But this doesn't seem to be the case in the tables reported by authors. Do they consider multilingual inference? Some clarifications would help.

-  Moreover, if the authors wanted to reinforce their argument about efficiency, only counting the amount of the parameters doesn't give the full picture (parameter count only address memory efficiency): inference speed, the amount of flops (in training and inference) would provide additional and important view to get a full picture.

- I am not sure the choice of CLSR as a baseline. There are many PEFT methods these days that could serve as a reasonable baseline for such approach (eg. laguage-specific or language-pair specific adapter layers seems like a very natural baseline to me, and should probably be comparable in terms of parameter efficiency )

**Reproducibility:**

5: Could easily reproduce the results.

**Reviewer Confidence:**

5: Positive that my evaluation is correct. I read the paper very carefully and I am very familiar with related work.

---

> ### Author Rebuttal · Authors · 2023-08-29
>
> We genuinely appreciate the valuable feedback provided by the reviewer and have addressed them in a point-by-point manner below. We are more than willing to engage in further discussions with the reviewers should any follow-up questions arise.
>
> ### **Regarding your concern about reproducibility**
> >do you strat from pretrained model and add LMS layers on top of it or do you train everything from scratch?
>
> Thanks for the detailed question! We start training everything from scratch! We note that this is in contrast to other work in the field and is one of the novel aspects of our proposed method.
>
> >Do LMS-FD-* models start from LMS model or do you train LS and shared modules simultaneously?
>
> For the LMS-FD-* models, we simultaneously train the LS and shared modules, as delineated in Equation 3. We appreciate the feedback and will provide a clearer elaboration on this in the revised manuscript. **Regarding the other hyperparameters, they are comprehensively detailed in Appendix A and D to facilitate reproduction**. Furthermore, we plan to release the code following the final decision on the paper.
>
> ### **Regarding your concern about computing inference parameters**
> >I do not have full understanding about how do authors compute inference parameters.
>
> We appreciate your insightful feedback. We agree that the number of parameters required by LMS-FD-LS is equivalent to those of LMS-FD-Share if and only if evaluated in a single translation direction. Nonetheless, our work involves translating in multiple directions. Therefore, we would like to define the “number of inference parameters used by LMS-FD-*” as the total number of parameters needed to complete all required translations. In practical scenarios, this refers to the number of parameters loaded into memory at a given time, because it is impractical to reload a model for each translation direction due to the time cost and its infeasibility for production. We will clarify this in the paper.
>
> ### **Regarding your concern about full picture of efficiency**
> >inference speed, the amount of flops (in training and inference) would provide additional and important view to get a full picture.
>
> We appreciate your valuable suggestions. For clarity, we want to remind the reviewer that we discussed inference speed and Flops/tok in the Limitations section (lines 569-583). In light of your feedback, we will highlight it on the main pages with an extra space. It's worth noting that our training and inference speed aligns closely with that of the naive MMT and top-1 MoE training (Please see detailed numbers in “**Regarding your question 4**” below). Moreover, our method boasts a reduced parameter count and enhanced performance.
>
> ### **Regarding your concern about CLSR and PEFT baselines**
> >I am not sure the choice of CLSR as a baseline. There are many PEFT methods these days that could serve as a reasonable baseline for such approach
>
> We are grateful for your constructive feedback. It's important to highlight that the central aim of our paper is to introduce lightweight language-specific modules that counteract negative language interference, **specifically in models that are trained from scratch**. While CLSR aligns with our goal and employs an adaptive routing algorithm to LS modules (which we view as a standard baseline), PEFT is tailored for fine-tuning pre-trained models, serving a different purpose. While we strive to maintain clarity and brevity by not delving deeply into yet-unproposed baselines (e.g., LS prefix tuning in MMT and models entirely trained from scratch), we prioritize existing baselines that resonate with our core goals. Nevertheless, we do explore potential integrations of PEFT and LS modules when training from scratch:
>
> * LoRA: LoRA closely mirrors our LS component, employing two low-rank matrices to forge a larger matrix. In essence, the composition of language-wise LMS can be perceived as a parallel LoRA. For a more comprehensive understanding, we've included additional experiments focused on the conventional language-specific sequential LoRA, ensuring they too are trained from scratch in alignment with LMS. The training and evaluation dataset is IWSLT’14 dataset.
>
> |   Methods  |     ar    |     de    |     es    |     fa    |     he    |     it    |     nl    |     pl    |    avg.   |
> |:----------:|:---------:|:---------:|:---------:|:---------:|:---------:|:---------:|:---------:|:---------:|:---------:|
> | Naive MMT  |   25.03   |   32.59   |   39.98   |   18.76   |   33.39   |   34.00   |   36.71   |   22.37   |   30.35   |
> | LS-LoRA    |   +0.32   |   +0.15   |   +0.32   |   +0.38   |   +0.64   |   +0.44   |   +0.12   |   +0.74   |   +0.40   |
> | LMS (ours) | **+0.87** | **+1.08** | **+1.04** | **+0.62** | **+1.37** | **+1.20** | **+1.04** | **+1.16** | **+1.05** |
>
> * Prompt tuning and prefix tuning: In our MMT training approach, a special token is typically prefixed to source sentences, signaling the target language. This process in naive MMT can be equated to a length-1 LS prompt tuning. Prefix tuning bears a resemblance to prompt tuning. However, the LS parameters in these methods are often too limited to encapsulate translation or multilingual information, making them less prevalent in translation tasks.
>
> ### **Regarding your question 1:**
> >Does this mean that you do not initatialize model LMS-FD-LS from LMS, but train it from scratch?
>
> We appreciate the reviewer's questions! Indeed, the LMS-FD-LS is not initialized from LMS; both are trained from scratch. Given that LMS-FD incorporates an additional distillation loss and undergoes another pass, we deemed it appropriate to train LMS-FD from scratch as well. Notably, LMS-FD possesses an additional (language-shared) low-rank matrix compared to LMS. Consequently, the training strategy for LMS-FD encompasses regular MT training complemented by distillation between the LS low-rank matrix and the shared low-rank matrix.
>
> ### **Regarding your question 2**
> >I do not understand what does WR(%) refers to in tables 3 and 7
>
> Please allow us to clarify it more clearly: WR is the proportion of language pairs on which our method beats the baseline, as we mentioned in lines 358-360. For example, if we beat the baseline 99 out of 100 languages, the WR is 99%. We appreciate the feedback and will provide a clearer elaboration on this in the revised manuscript.
>
> ### **Regarding your question 3**
> >I am not sure I understand how do you compute training/inference parameters
>
> Thanks for your questions!  Please refer to our answers in “Regarding your concern in computing inference parameters
>
> ### **Regarding your question 4**
> >I think it would be useful to report total amount of flops in training and inference.
>
> Thanks for your suggestions!  Please refer to our answers in “Regarding your concern in full picture of efficiency”. We previously noted that the FLOPs/tok and time costs are similar to those of the naive MMT and top-1 MoE. For a comprehensive understanding, we provide a detailed breakdown of the FLOPs/tok for all experiments (primary results in OPUS-100) in Table 3:
>
> | Methods            | en->xx avg. | xx->en avg. | FLOPs/tok, train | FLOPs/tok, infer |
> |--------------------|:-----------:|:-----------:|:----------------:|:----------------:|
> | Naive MMT          |    27.37    |    30.60    |       506M       |       506M       |
> | Switch transformer |    +2.66    |    +0.84    |       506M       |       506M       |
> | CLSR               |    +0.02    |    +1.83    |       556M       |       556M       |
> | LMS, d=64          |    +3.60    |    +3.35    |      507.6M      |      507.6M      |
> | LMS-FD-Share,d=64  |    +0.74    |    +1.22    |      507.6M      |      507.6M      |
> | LMS-FD-LS, d=64    |    +2.01    |    +2.52    |      507.6M      |      507.6M      |
> | LMS, d=16          |    +2.53    |    +2.39    |      506.4M      |      506.4M      |
> | LMS-FD-Share, d=16 |    +1.09    |    +1.17    |      506.4M      |      506.4M      |
> | LMS-FD-LS, d=16    |    +1.77    |    +2.10    |      506.4M      |      506.4M      |
> | LMS, d=4           |    +1.95    |    +1.55    |      506.1M      |      506.1M      |
>
> ### **Regarding your question 5**
> >it would be interesting to report full results decomposed per language pair somewhere (in the appendix, or on separate document) for better
>
> We deeply appreciate your insightful suggestions. To address this, we will include the complete results for each language pair in the appendix of the camera-ready version.

---

### Official Review · Reviewer_Khj5 · 2023-08-04

**Soundness:** 3

**Excitement:**

3: Ambivalent: It has merits (e.g., it reports state-of-the-art results, the idea is nice), but there are key weaknesses (e.g., it describes incremental work), and it can significantly benefit from another round of revision. However, I won't object to accepting it if my co-reviewers champion it.

**Paper Topic And Main Contributions:**

In this work, the authors propose a method similar to LoRA (under a different name) on the MLPs of transformer models to build language-specific modules. They also propose a method to distill these language-specific modules into a single, shared module to save model size. They then evaluate their methods on 3 tasks: NER, MT, and QA.

**Questions For The Authors:**

A) In line 139 you define W_{l_i -> l_j} = W_v^{l_i} W_f{l_j}. Why not the reverse other, ie, W_v{l_j} W_f^{l_i}? If the idea is to go from language i to language j, then presumably the input is in language i and the order I proposed seems more "natural". Did you try this alternative? If not, why?
A.1) Slightly related: you never state whether you use row or column vectors. Looking at equation 1, it seems you use the latter, but you should state this clearly.
B) Regarding equation 2, I assume you compute it just at the output layer (and not at each layer), right?
B.1) If you're using g(.) as a stop_gradient function, shouldn't the formula be: KL(g(p_l) || p_s) instead?
C) On equation 3, isn't the first term frozen during the fuse distillation? What is trained and what is fixed? It is not entirely clear from the paper where you're updating the weights at each point in time.
D) In Table 1 you show the naive model, FD, and FD-LS. Where is just LS? This question applies to the rest of the paper. You never show LS alone, but always FD-LS.
E) During FD-LS, are the LS weights trained, or just the shared weights? I assume they are, because you say in section 3.2 that "during the inference stage the input is processed through its LS FFN module", and it would make sense otherwise.
F) In equation 4, you propose using the Jensen-Shannon divergence (I'm pretty sure you can find some older paper to cite rather than Jiang et al 2020). What is the reason for this? Did you try the regular KL divergence and it didn't work?
G) In lines 344-345 you say you "fix the training step for all methods for a fair comparison". Are all models trained to convergence? Otherwise I'm not convinced the comparison is fair.
I) In line 451 you say "we set d=32". Similar on line 497, "d=20". How did you pick this number?
J) I don't think you have enough evidence to say on lines 457-459 that "FD may be more effective in fusing knowledge when the number of languages is relatively small", since you're comparing different tasks. Is there other evidence?

**Reasons To Accept:**

* The idea of using LoRA to tune the FFNs to be language specific is interesting. The proposed approach for distillation is also interesting.

**Reasons To Reject:**

* The authors call their method "Language-Specific Matrix Synthesis", but their method is essentially LoRA for multilingual settings. I still think it's interesting, but this attempt to make it sound more novel than it really is is a negative point in my opinion.
* The paper claims their method "significantly outperforms previous language-specific or mixture of experts methods" (lines 77-78), yet they don't compare with any of those methods. There is plenty of related work that is cited on section 7, but no comparisons were made. While it is interesting that the authors test their method on different tasks, the lack of any meaningful baseline is a serious omission.

I do think the idea is interesting and that this can be an interesting paper, but as it stands I strongly believe it would benefit from further work.

**Reproducibility:**

3: Could reproduce the results with some difficulty. The settings of parameters are underspecified or subjectively determined; the training/evaluation data are not widely available.

**Reviewer Confidence:**

4: Quite sure. I tried to check the important points carefully. It's unlikely, though conceivable, that I missed something that should affect my ratings.

**Typos Grammar Style And Presentation Improvements:**

* Line 40: "becasue" -> "because".
* Line 109: "martices" -> "matrices".
* Line 193: add "an existing model with" after "shared FFN parallel to".
* Line 263: "need" -> "needed".
* Line 280: "Equation 3" -> "Equation 4"?
* In the text (page 7) you refer to the pair-wise results on Table 3, but the table doesn't mention which results are pair-wise. I suppose it's the 5th result from the top?
* Line 344: "training step" -> "training steps".
* Line 883: "Figure 1" -> "Table 9".
* Table 9 caption: "for the Figure 1" -> "for Figure 1".

---

> ### Author Rebuttal · Authors · 2023-08-28
>
> We genuinely appreciate the valuable feedback provided by the reviewer and have addressed them in a point-by-point manner below. We are more than willing to engage in further discussions with the reviewers should any follow-up questions arise.
>
> ### **Regarding your concern about the Method name**:
> >The authors call their method "Language-Specific Matrix Synthesis", but their method is essentially LoRA for multilingual settings.
>
> We are grateful for your feedback and would like to provide further clarification. Firstly, LoRA and LMS differ fundamentally in their approaches. While LoRA serves as a general lightweight fine-tuning method for pre-trained models, LMS ( and the whole model) is explicitly trained from scratch to encapsulate LS information. Secondly, we acknowledge and value the contributions of LoRA. As evidence, we have cited the foundational paper for LoRA (line 178) as well as the work on LoRA itself (line 533). It's essential to note that LMS introduces specific nuances and modifications tailored for multilingual contexts, making it more than a mere variant of LoRA. Recognizing the need for transparency and clear articulation, we commit to ensuring our revised manuscript accurately depicts the relationship between our method and LoRA, highlighting the unique aspects of our approach without any unintentional overstatement.
>
> ### **Regarding your concern about Baselines**
> >The paper claims their method "significantly outperforms previous language-specific or mixture of experts methods" (lines 77-78), yet they don't compare with any of those methods.
>
> We deeply appreciate your insightful feedback! Both the CLSR and our method aim towards a shared goal: leveraging language-specific parameters to counteract negative language interference. The CLSR stands as a robust, widely-validated, and foundational method in language-specific models, rendering it a quintessential baseline in our view. Similarly, the Switch transformer serves as a foundational Mixture-of-Experts (MoE) model upon which many MoE models are structured. We respectfully disagree with the point that the chosen baselines are meaningless. In Section 7, we extensively reference related work, underscoring our regard for their contributions. Admittedly, drawing comparisons across every method would be challenging, leading us to opt for two standard methods as baselines.
>
> Nonetheless, to underscore the prowess of our approach and follow the reviewer’s suggestion, we benchmark our model against two cutting-edge MoE models: NLLB-EOM and NLLB-CMR [1]. Notably, both these models amplify the computational burden at the FFN layer due to their utilization of top-2 gating, with NLLB-CMR also leveraging one more FFN layer during training. Our LMS method, in contrast, outperforms these state-of-the-art MoE techniques while consuming fewer computational resources and parameters:
> | Methods            | en-xx, high  | en-xx, med | en-xx, low | en-xx, all | xx-en, high | xx-en, med | xx-en, low | xx-en, all |
> |--------------------|:------------:|:----------:|:----------:|:----------:|:-----------:|:----------:|:----------:|:----------:|
> | Naive MMT          |     23.89    |    31.17   |    29.76   |    27.37   |    29.40    |    31.85   |    31.49   |    30.60   |
> | Switch transformer |     +1.87    |    +3.29   |    +3.51   |    +2.66   |    +1.18    |    +1.15   |    -0.31   |    +0.84   |
> | NLLB-EOM           |     +2.35    |    +3.59   |  **+4.1**  |    +3.11   |    +1.83    |    +1.84   |    +1.95   |    +1.85   |
> | NLLB-CMR           |     +2.43    |    +3.81   |    +4.01   |    +3.19   |    +1.93    |    +1.84   |    +1.62   |    +1.83   |
> | LMS, d=64 (ours)   |   **+3.60**  |  **+3.82** |    +3.32   |  **+3.60** |  **+2.75**  |  **+3.74** |  **+4.16** |  **+3.35** |
>
> Reference:
>
> [1] NLLB Team, Costa-jussà MR, Cross J, Çelebi O, Elbayad M, Heafield K, Heffernan K, Kalbassi E, Lam J, Licht D, Maillard J, Sun A. No language left behind: Scaling human-centered machine translation. arXiv preprint arXiv:2207.04672. 2022 Jul 11.
>
> ### **Regarding your question A**
> >Why not the reverse other, ie, W_v{l_j} W_f^{l_i}?
>
> Thanks for your interesting suggestions!  The primary difference that the reverse order $W_v{l_j}W_f^{l_i}$ will make is that the matrix originally serving the language direction $i\rightarrow j$ will serve $j\rightarrow i$. We would expect that they have a similar performance, so we do not further dig into the reverse order.
>
> ### **Regarding your question B**
> >I assume you compute it just at the output layer (and not at each layer), right?
>
> Yes, we are computing at the output (final) layer rather than each layer. The outputs of the middle layers are vectors that cannot be used by computing KL divergence. We appreciate for pointing out the ambiguity and will provide a clearer elaboration on this in the revised manuscript.
>
> >If you're using g(.) as a stop_gradient function, shouldn't the formula be: KL(g(p_l) || p_s) instead
>
> Thanks for pointing this out! Yes, the stop gradient function should be on the left side as you indicated!
>
> ### **Regarding your question C**
> >On equation 3, isn't the first term frozen during the fuse distillation? What is trained and what is fixed?
>
> We appreciate the opportunity to clarify the components of Equation 3. Within the equation, the first two items in the bracket $(CE(y ∥ p_l)$ and $CE(y ∥ p_s)$ are not frozen because they are regular machine translation losses to learn how to translate. We also need to learn the original task while distilling the model. The third item $L_{fd}$ is used for learning from LS components to shared components as we call fuse distillation. All terms in Equation 3 are trained concurrently.
>
> ### **Regarding your question D**
> >In Table 1 you show the naive model, FD, and FD-LS. Where is just LS?
>
> We appreciate your insightful suggestion. The primary focus of Section 3 is to elucidate the feasibility of condensing multilingual knowledge into a shared module, thus the concise presentation. However, for clarity and completeness, we present the LS results. It can be observed that our FD approach performs similarly to (even slightly outperforms) the basic LS method.
>
> |  Methods  |    ar    |     de    |     es    |     fa    |     he    |     it    |     nl    |     pl    |    avg.   |
> |:---------:|:--------:|:---------:|:---------:|:---------:|:---------:|:---------:|:---------:|:---------:|:---------:|
> | Naive MMT |   25.03  |   32.59   |   39.98   |   18.76   |   33.39   |   34.00   |   36.71   |   22.37   |   30.35   |
> | LS        |   +1.06  |   +1.12   |   +1.43   |   +0.87   |   +1.10   |   +1.11   |   +1.28   |   +1.02   |   +1.13   |
> | FD        |   +1.01  |   +1.15   |   +1.43   |   +0.64   |   +1.44   |   +1.19   |   +1.22   |   +1.22   |   +1.17   |
> | FD-LS     | **+1.3** | **+1.45** | **+1.72** | **+0.77** | **+2.08** | **+1.48** | **+1.41** | **+1.73** | **+1.50** |
>
> ### **Regarding your question E**
> >During FD-LS, are the LS weights trained, or just the shared weights?
>
> We appreciate the opportunity to provide further clarification. Both LS and shared weights are trained.  FD is trained by two forward passes, one with LS modules and one with shared modules, and by one backward pass with regular MT loss and a distillation loss. As indicated in line 230, FD-LS is exactly the model trained by FD method, but the only difference is how to inference. Evaluation of FD only utilizes shared modules but evaluation of FD-LS only uses LS modules.
>
> ### **Regarding your question F**
> >you propose using the Jensen-Shannon divergence (I'm pretty sure you can find some older paper to cite rather than Jiang et al 2020). you can find some older paper to cite rather than Jiang et al 2020). What is the reason for this?
>
> Please allow us to correct that symmetric KL divergence is not the same as the Jensen-Shannon divergence. The definition of JS divergence is $JS(p|q)=\frac{1}{2} (KL(p|m) + KL(q|m))$, where $m=\frac{p+q}{2}$. However, symmetric $KL=\frac{1}{2}*(KL(p|q) + KL(q|p))$. The pioneering work by SMART (Jiang et al 2020) [1] introduced the symmetric KL and employed it within the smoothness-inducing adversarial regularizer, a process necessitating two forward passes. Following in their footsteps, previous studies like R3F [2] and THOR [3] that utilize two passes have also cited SMART for their adoption of symmetric KL.  We wish to duly acknowledge and honor these foundational contributions, hence our citation of the work.
>
> >Did you try the regular KL divergence and it didn't work?
>
> Thanks for your interesting question. We appreciate the insightful query. While we have not experimented with the standard KL divergence, we anticipate its potential efficacy. Given the common parameter $W$ involves both LS and shared paths, our intention was to learn $W$ symmetrically, leading us to opt for symmetric KL over the conventional KL, primarily due to the asymmetry inherent in the latter. Moreover, as illuminated by preceding works such as [1,2,3], symmetric KL has demonstrated superior performance over simple KL in similar contexts.
>
> Reference:
>
> [1] Jiang H, He P, Chen W, Liu X, Gao J, Zhao T. SMART: Robust and Efficient Fine-Tuning for Pre-trained Natural Language Models through Principled Regularized Optimization. InProceedings of the 58th Annual Meeting of the Association for Computational Linguistics 2020 Jul (pp. 2177-2190).
>
> [2] Aghajanyan A, Shrivastava A, Gupta A, Goyal N, Zettlemoyer L, Gupta S. Better fine-tuning by reducing representational collapse. arXiv preprint arXiv:2008.03156. 2020 Aug 6.
>
> [3] Zuo S, Liu X, Jiao J, Kim YJ, Hassan H, Zhang R, Gao J, Zhao T. Taming Sparsely Activated Transformer with Stochastic Experts. International Conference on Learning Representations 2021 Oct 6.
>
> ### **Regarding your question G**
> >"fix the training step for all methods for a fair comparison". Are all models trained to convergence?
>
> We appreciate the opportunity to provide further clarity on this matter. Employing a fixed number of training steps is a standard approach in multilingual machine translation, as evidenced by a plethora of prior studies, including [1,2,3,4] (see reference below). We opted for this approach for several reasons:
> * Scale of Model and Data: Given the substantial size of our model and data, pinpointing the exact moment of "convergence" based on the rise of evaluation loss becomes challenging. Specifically, our most extensive model encompasses 1B+ parameters, coupled with a data size of 55M. Training such a model for weeks to discern convergence signals would be impractical.  Nevertheless, we've designated a rational 100K training step, which has shown to be sufficient for the model to deliver robust performance [2]. Importantly, we apply this same constraint across all models, ensuring an equitable comparison, as also advocated by preceding works.
> * Composite Evaluation Loss: Multilingual machine translation inherently involves multi-task learning, where the evaluation loss aggregates results from all the 100 languages. Continuing training may lead performance on some languages to improve while others may degrade and get worse,  making it challenging to determine a universally "converged" point.
> * Reasonable settings: Based on empirical evidence and guidelines from previous works like [2], we've configured our training settings — 100K steps for a 65K batch size and 150K steps for a 4K batch size in OPUS and IWSLT datasets, respectively. These settings have been verified to achieve robust and good results.
>
> Reference:
>
> [1] Zhang B, Williams P, Titov I, Sennrich R. Improving massively multilingual neural machine translation and zero-shot translation. arXiv preprint arXiv:2004.11867. 2020 Apr 24.
>
> [2] NLLB Team, Costa-jussà MR, Cross J, Çelebi O, Elbayad M, Heafield K, Heffernan K, Kalbassi E, Lam J, Licht D, Maillard J, Sun A. No language left behind: Scaling human-centered machine translation. arXiv preprint arXiv:2207.04672. 2022 Jul 11.
>
> [3] Xu H, Maillard J, Goswami V. Language-Aware Multilingual Machine Translation with Self-Supervised Learning. arXiv preprint arXiv:2302.05008. 2023 Feb 10.
>
> [4] Kim YJ, Awan AA, Muzio A, Salinas AF, Lu L, Hendy A, Rajbhandari S, He Y, Awadalla HH. Scalable and efficient moe training for multitask multilingual models. arXiv preprint arXiv:2109.10465. 2021 Sep 22.
>
> ### **Regarding your question I**
> > "we set d=32". Similar on line 497, "d=20". How did you pick this number?
>
> Thank you for your insightful feedback. These numbers were selected based on empirical observations from our primary MMT tasks,  where these values were observed to introduce only a minimal increase in parameters while maintaining substantial improvements. Although we acknowledge that these hyperparameters were not exhaustively tuned, they serve to ensure a consistent and meaningful comparison. For instance, both the LMS with ffn-only and attn-only utilize the same d=20.
>
> ### **Regarding your question J**
> >I don't think you have enough evidence to say on lines 457-459 that "FD may be more effective in fusing knowledge when the number of languages is relatively small", since you're comparing different tasks. Is there other evidence?
>
> Thank you for the question. We do have other evidence and will clarify it more succinctly in the paper. When exploring multilingual machine translation in the context of a 15-language setup (refer to Table 2), the performance disparity between LMS-FD-Share and LMS-FD-LS is notably narrower (+0.88 vs. +1.26) in comparison to the 100-language configuration presented in Table 3 (+0.74 vs. +3.60 with rank=64). This observation suggests that within the 15-language framework, the information encapsulated within the shared module aligns more closely with that of the LS modules. Moreover, intuitively, with fewer languages in play, the FD approach should find it easier to encapsulate less multilingual knowledge.

---

### Official Review · Reviewer_fwLV · 2023-08-09

**Soundness:** 5

**Excitement:**

4: Strong: This paper deepens the understanding of some phenomenon or lowers the barriers to an existing research direction.

**Paper Topic And Main Contributions:**

The paper is proposing a new approach for multilingual translation by introducing language specific module, which achieves the state-of-the-art in the multilingual benchmark. Aside from the superior performance, it has a better scalability when it is compared with other architecture such as SwitchFormer, even if they both use a mixture of sub-modules.

**Questions For The Authors:**

* I am curious to see the comparison of this approach with other computationally-light methods such as Adapter https://github.com/adapter-hub/adapter-transformers



**Reasons To Accept:**

The idea is simple yet smart and intuitive, which is supported by the strong performance in the desired downstream task. Beside the performance, it scales to a number of languages, and even propose a technique to pack all the knowledge into a single module, to further reduce the complexity. It's well grounded and solid study.

**Reasons To Reject:**

I'm mostly convinced that the paper should get accepted, as there is nothing to complain.

**Reproducibility:**

3: Could reproduce the results with some difficulty. The settings of parameters are underspecified or subjectively determined; the training/evaluation data are not widely available.

**Reviewer Confidence:**

3: Pretty sure, but there's a chance I missed something. Although I have a good feel for this area in general, I did not carefully check the paper's details, e.g., the math, experimental design, or novelty.

---

> ### Author Rebuttal · Authors · 2023-08-28
>
> We greatly appreciate your recognition and positive support!
>
> >I am curious to see the comparison of this approach with other computationally-light methods such as Adapter https://github.com/adapter-hub/adapter-transformers
>
> We greatly appreciate your recognition and positive support! Thanks for your suggestions! For a more comprehensive understanding, we've included additional experiments focused on the conventional language-specific (LS) sequential LoRA, ensuring they too are trained from scratch in alignment with LMS. The training and evaluation dataset is IWSLT’14 dataset. Our method also substantially outperforms LS-LoRA.
>
> |   Methods  |     ar    |     de    |     es    |     fa    |     he    |     it    |     nl    |     pl    |    avg.   |
> |:----------:|:---------:|:---------:|:---------:|:---------:|:---------:|:---------:|:---------:|:---------:|:---------:|
> |  Naive MMT |   25.03   |   32.59   |   39.98   |   18.76   |   33.39   |   34.00   |   36.71   |   22.37   |   30.35   |
> |   LS-LoRA  |   +0.32   |   +0.15   |   +0.32   |   +0.38   |   +0.64   |   +0.44   |   +0.12   |   +0.74   |   +0.40   |
> | LMS (ours) | **+0.87** | **+1.08** | **+1.04** | **+0.62** | **+1.37** | **+1.20** | **+1.04** | **+1.16** | **+1.05** |

---

### Meta-Review · Area_Chair_17iS · 2023-09-19

**Recommendation:** 5

**Metareview:**

This paper presents a new way to enhance the efficiency of multilingual neural machine translation (MNMT). First, it introduces an efficient alternative to language adapters, termed Language Specific Matrix Synthesis (LMS). Second, it presents Fuse Distillation (FD), which aims to distill all the knowledge encapsulated in language-specific modules into a single shared module. The experimental results show that LMS helps reduce the parameter count without compromising performance. Furthermore, FD reduces the parameter count while also improving MNMT, as it essentially serves as a regularization method.

Pros:
- I agree with the reviewers regarding this work's innovative approach to achieving both efficiency and effectiveness in MNMT. The concepts presented are intriguing, and the results demonstrate notable improvements in both performance and efficiency.
- The paper is well written, making it easy to comprehend and potentially straightforward to replicate. It includes comprehensive and thorough analyses.

Cons:
- While there are minor details that require refinement in the final version, they have all been addressed well in the rebuttal phase.

---

### Decision · Program_Chairs · 2023-10-07

**Decision:**

Accept-Main

**Comment:**

This paper presents a new way to enhance the efficiency of multilingual neural machine translation (MNMT). First, it introduces an efficient alternative to language adapters, termed Language Specific Matrix Synthesis (LMS). Second, it presents Fuse Distillation (FD), which aims to distill all the knowledge encapsulated in language-specific modules into a single shared module. The experimental results show that LMS helps reduce the parameter count without compromising performance. Furthermore, FD reduces the parameter count while also improving MNMT, as it essentially serves as a regularization method.

Pros:
- I agree with the reviewers regarding this work's innovative approach to achieving both efficiency and effectiveness in MNMT. The concepts presented are intriguing, and the results demonstrate notable improvements in both performance and efficiency.
- The paper is well written, making it easy to comprehend and potentially straightforward to replicate. It includes comprehensive and thorough analyses.

Cons:
- While there are minor details that require refinement in the final version, they have all been addressed well in the rebuttal phase.